

# Geo-Informatics Data Acquisition Instrument for Assessment of Avalanche Risk and Trafficability: STMET

Ganesh Kumar, Zorawar Singh

Defence Geoinformatics Research Establishment (DGRE), Him Parisar, Sector- 37, Chandigarh, PIN-160036, India

*Correspondence to: Ganesh Kumar(ganesh.kumar.dgre@gov.in)*

**Abstract.** Off-road soil conditions and snowpack strata data are traditionally acquired with Cone penetrometers and Ramsonde Rod to assess trafficability and snow stability. Snow and Terrain Mobility Evaluation Tool (STMET) has been designed and developed as a multi-utility instrument for snow and soil to acquire geo-informatics data. Load cell, moisture sensor, temperature sensor, and data transmission modules in small form factors are embedded in the rod-like instrument for easy movement and for

finding penetration resistance, moisture, and temperature profile of snowpack and soil.  One of the Cone angles (30°, 45° and 60°) of the cone assembly is auto-detected by the data acquisition system for ease of set-up and data storage depending on geophysical conditions. Laser-based ranging sensors continuously monitor cone penetration depth. The profiles of snow strength, moisture and temperature are displayed on the detachable display in graphical form which helps to find the weak layer responsible for avalanche occurrences. The position of the experimental site and the terrain slope are obtained with the Global Navigation Satellite System

(GNSS) and Inertial Measurement Unit (IMU) built into the system. The transmission module of the system transmits the data and an alert signal to a distance in case of an emergency. The catchment area of an avalanche site is measured using an algorithm based on the data obtained with a distance-measuring sensor and Roll-Pitch-Yaw data of IMU. The enhanced capability of the instrument for trafficability assessment is feasible by comparing the soil cone index (CI) and Vehicle Cone Index (VCI). The Decision Support System (DSS) has been implemented as a tool for soil, snow stratigraphy, avalanche and victim detection to ease the decision-

making process for assessing trafficability and avalanche risk. This paper highlights the concept and features of the Snow and Terrain Mobility Evaluation Tool (STMET) capable of acquiring geo-informatics data for scientific applications in difficult terrains.

## 1. Introduction

Trafficability is the ability of terrain conditions to support vehicle movement without any contingency mechanical effort(He et al.,

2020). It mainly depends on microscopic soil characteristics (compactness, bearing capacity and moisture), and snow and macroscopic terrain properties (vegetation, slope and roughness). Many researchers have tried to establish the relationship between off-road trafficability and soil properties (Knight and Freitag, 1961; Taheri et al., 2015). The cone penetrometer provides soil strength based on soil characteristics and it is found in terms of cone index (CI). It is compared with the Vehicle Cone Index (VCI) to assess trafficability (Taheri et al., 2015). Complementing this method, various other methods such as rut measurement techniques

have also been undertaken and developed. Bevameter is used to estimate the shear strength of the soil for trafficability assessment (Wong, 2001). Moisture of soil is one of the important parameters for trafficability(Pytka et al., 2019; Smith, 1973).  However, neither a cone penetrometer nor a Bevameter has an arrangement to measure soil moisture and temperature which have a tremendous impact on the terrain characteristics and trafficability(Perumpral, 1983). Therefore, a provision for the measurement of these vital parameters has been included in the present tool by incorporating moisture and temperature sensors.




Snow depositions are built up in different layers after several precipitations and the properties of these snow layers are changed due to the metamorphism process(Irwin, 1980). The weak layer present in the formation zone of the snowpack destabilises to trigger a slab avalanche (McClung and Schaerer, 1993; Schweizer et al., 2003). Avalanches greatly threaten human lives and infrastructure (McClung and Peter, 1999). It causes problems in the movement of defence personnel and the civilian population in these areas.

In the past century, various methods and models have been developed and assessed to obtain the actual characteristics of terrain like Bekker's model (1969), Brixius equations (1987), Reece's model (1965), Lyasko analytical model (2010), FEM and DEM methods (He et al., 2019). The work paves the way for a new field of study, Terrain-Vehicle Systems (He et al., 2020; Kennedy and Rush, 1968; Rush E.S. and Robinson J.H., 1971). The practical importance of constructing such models is based on the fact that numerous types of terrains and vehicles exist, resulting in countless combinations of terrain-vehicle interactions. These

interactions need to be assessed from the preview of the scientific hypothesis, vehicle performance, effectiveness, traversability predictions for new machines and routes and cost. A well-known development in this field is the Next Generation NATO Reference Mobility Model (NG-NRMM) (NG-NRMM Final Report, Technical Activity Proposal 2014)and autonomous vehicles for Off-road mobility (Islam et al., 2022). Generally, the Cone index is used as a measure of soil strength for terrain assessment in soil mechanics and the humidity level is used to measure the moisture of the soil to check root impedance but the same is not true for

snow covers (Perumpral, 1983).

Terrain characterization methods require technically refined tools and a team of well-trained geo-technical personnel to extract useful data to be keyed into the terra mechanics models. Similarly, snow stratigraphy requires experienced and scientifically trained snow hydrologists to extract the data for decision-making. An instrumented avalanche rod (IAR) provides snowpack information and has the provision of attachment safety equipment (Chasmai et al., 2012). IAR is used in the Himalayan region but

has limited applications. Snow Micro Pen (SMP) (Techel et al., 2008) is a motorized probe that has miniature load cells at its tip to measure penetration resistance and to know the snow strength profile of the snowpack. The hardness of the snowpack is measured with a Ramsonde rod during stratigraphy, it is conducted out for avalanche forecast. Thus different tools and instruments are being used to measure different parameters of the snow to assess the stability of the snow pack and reduce the avalanche risk. These instruments are specially designed for snow and cold temperatures. There has been a requirement for an instrument which

can work in soil and snow for trafficability as well assessing the stability of snowpack. Keeping these requirements in mind, the Snow and Terrain Mobility Evaluation Tool (STMET) has been designed and tested in the field. Snow and Terrain Mobility Tool (STMET) is a multi-utility instrument to be used in snow as well as soil terrain for assessment of snow stability and trafficability of off-road vehicles. A Decision Support System (DSS) based on geo-informatics data acquired with the instrument has been incorporated into it for avalanche safety and trafficability. In this paper, the design of the instrument, its data processing system

and its methodology are elaborated.



## 2. Design and Techniques of Geo-Informatics Parameter Measurement

### 2.1 Design of Instrument

The Snow and Terrain Mobility Evaluation Tool (STMET) is designed to fulfil the requirement of a versatile instrument, workable in snow and different soils to acquire the geophysical parameters. It is made in the form of a stick to facilitate easy-to-walk with it

in difficult terrain and simultaneously to acquire geo-informatics data along the route. Its portability and lightweight (less than 2 kg) features are intended to make it easy to carry for field experimentation. It houses sensors within a small form factor with a power source packed inside the handle. The working of the sensors is explained in a subsequent paragraph.

The different parts and the positions of the sensors of the instrument are depicted in Fig. 1.

Handle for Holding

Hand Press for Load application

Electronics Box containing load cell, range finder, depth measurement sensor, IMU, GPS

Multiple Rod Segments

Cone Assembly with temperature and humidity sensor

**Figure 1.** Snow and Terrain Mobility Evaluation Tool (Schematic Diagram)

To make the rod modular and extendable up to 4m, there is a provision for the addition of intermediate segments while maintaining signal continuity and data transmission. PCB, load cell, depth sensor and range finder have been enclosed in a specially designed box made of nylon. Nylon is an easily available lightweight material with sufficient strength and thermo-electric insulator. The handgrip of the handle and electronic box are made of it. A detachable display system is used to monitor processed data and to make decisions in situ. Three cones 30°, 45° and 60° (Fig. 2) are provided with the instrument to be used in different terrain

conditions. PCBs of cones are configured for their auto-identification to avoid any human error during the experiments.

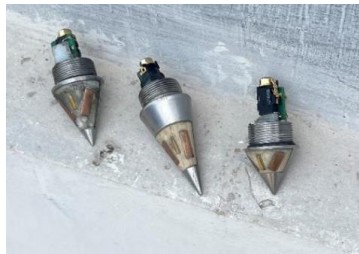

**Figure 2.** Cones of different angles



The design parameters of the instrument are given in Table 1.

**Table 1**. Design parameters of the instrument

| S. No. | Parameters | Specification |
|--------|-----------|---------------|
| 1 | Basic Length | 1.3 m |
| 2 | Extendable Length (up to) | 4 m |
| 3 | Material | Aluminium 6061 structure |
| 4 | Rod Diameter | 25 mm (OD) – 20mm (ID) |
| 5 | Working Temperature | -20 °C to +50 °C |
| 6 | Terrain type | Snow/Soil/Sand |
| 7 | Physical properties to be measured | Strength, moisture, Temperature |
| 8 | Distance range | 3 km |

**2.2 Geo-Informatics Parameters**

Geo-informatics data like soil condition and its strength, snow strength and temperature profile are essential for planning area development, avalanche safety measures and trafficability assessment. The importance and relevance of Geo-informatics data have increased with the migration of people towards snow-bound areas, growth in tourism and changing environmental conditions. To promote mitigation measures for geo-hazards like avalanches and landslides, it is important to assess the stability of snow mass in
the formation zone of avalanches and to know the moisture content of the soil. The stability criteria of snowpack depend on snow depth, snow strength, snow moisture, meteorological factors and terrain conditions. Similarly, the assessment of trafficability requires the input of soil strength and soil moisture. Taking into consideration the above requirements, STMET is used to acquire essential geo-informatics data. The geo-informatics parameters and their measuring technique have been described below.

**2.2.1 Depth Measurement**

The instrument is penetrated inside snow and soil manually to find their properties. The depth of penetration is measured using a laser sensor placed inside the electronic box facing towards the ground. The depth measurement is done using the principle of Time-of-Flight (ToF) up to a distance of 4m with a resolution of 1 mm which helps in identifying different layers of soil and snow. The laser sensor operates using an infrared wavelength. The rod of the tool is inserted through the hole made on a circular reflective plate and relative lengths are used to find the depth of the penetration. The depth of different layers is used to determine the weak
layer inside the snowpack. Similarly, the depth of soil is used to find the cone index profile for trafficability.

**2.2.2  Hardness Index Measurement**

The hardness of soil and snow is measured with Load cell assembly (Fig. 3) based on the resistance of soil felt by the rod. A load cell with a measurement range of 500 kg and resolution of 0.1 N has been fitted in the Electronics box which measures the load applied manually through the handle from the top. Depending on the terrain property, the reaction from the cone tip gets transferred
to the load cell. The hardness index is measured in terms of pressure which is converted to a soil cone index for the trafficability assessment with the help of customized software.



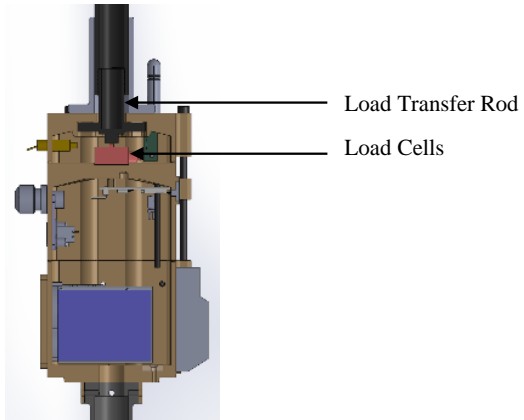

**Figure 3.** Load cell assembly

### 2.2.3 Temperature Profile Measurement

Snow is a thermodynamically unstable material so, the temperature of every layer of snowpack is important to predict the snow stability of avalanche paths. Similarly, the temperature in sub-surficial layers plays an important role in determining soil conditions. An NTC thermistor (Fig. 4) having a temperature range of - 40 °C to 50 °C with a resolution of 0.1 °C has been embedded at the bottom tip of the instrument. Its high resolution and fast response make it capable of recording the temperature profile of the snow layer of the snowpack. The temperature sensor along with depth sensors provides the snow temperature profile which is needed by
experts to forecast and predict snowpack stability.

### 2.2.4 Moisture Measurement

Moisture of snow and soil is one of the major parameters used for avalanche prediction and trafficability. It also helps in agriculture and soil tillage. The capacitance-based moisture sensor is developed and embedded in the metallic cone placed at the bottom part of the instrument. It has two copper strips (12mm x 4mm x 2mm) embedded parallel on the surface (Fig. 4).
The cone is inserted into the moist ground and water molecules change the permittivity of soil between the copper strips which in turn changes the capacitance between the strips as per Eq.1:

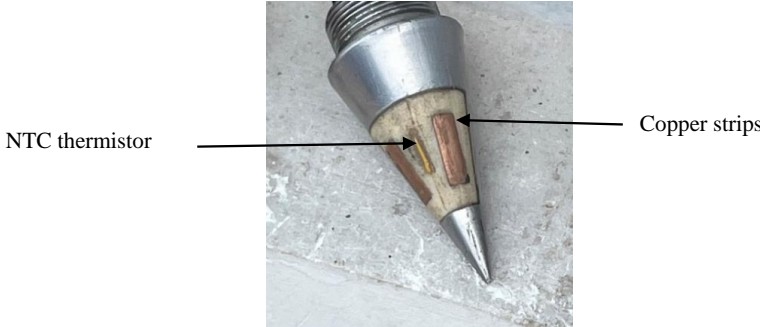

**Figure 4.** Capacitance-based moisture sensor and NTC Thermistor on Cone of the instrument



$$C = \frac{e_0 A}{d},$$ (1)

Where, $e_0$ is the dielectric constant of the space between the strips,

$A$ is the area of the strips,

$d$ is the distance between the strips.

The capacitance of the material varies concerning the percentage of water molecules polarization in the material. The average moisture of the material in contact with the copper strips was measured with this sensor. It has been calibrated with a known quantity of water (volumetric) in soil. Moisture of soil is also important in agriculture and STMET gives the moisture content of soil beneath the surface (Fig. 5).

**2.2.5 Ambient Temperature and Humidity**

Ambient temperature and humidity sensors provide instantaneous data for predicting, nowcasting and forecasting local weather conditions. These data would help the mountaineers while carrying the instrument and seriously wounded victims for the rescue operation. The values for ambient temperature and relative humidity are recorded with an Integrated Chip based sensor having a resolution of 0.01°C and accuracy of ±0.2°C for temperature readings and a resolution of 0.01 %RH and accuracy of ±1.8 %RH

capable of recording temperature ranges from -20° C to +50° C. The wide range of temperatures is crucial considering the application of the instrument in the snow and soil.

**2.2.6     Distance measurement**

The distance measurement is required to know the distance of objects, route planning, survey of the area and time management. A laser-based distance measurement sensor and telescopic sight are embedded with the instrument. It measures the distance of a far

object and helps to evaluate the area of the avalanche formation zone for avalanche hazard planning and rescue operations.

**3. Data Acquisition Process**

Snow and Terrain Mobility Evaluation Tool (STMET) is a multi-faceted instrument consisting of mechanical components, electronic sensors and a processing unit. STMET can be penetrated at any time easily, without snow being exposed to sunlight, to collect snowpack data with the help of its sensors attached at the bottom tip. More spatial data on snowpacks can be collected to provide

higher reliability in forecasting. The decision support system of the instrument helps to identify weak layers built inside the snowpack. The instrument can easily be used for snow data collection and safety in hazardous areas.

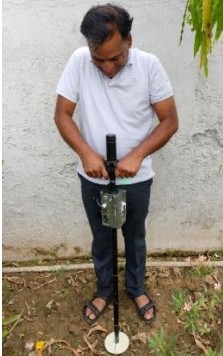

**Figure 5.** Data collection with STMET in field



The instrument is also used to determine the cone index of the soil which is further compared with the vehicle cone index (VCI) for assessment of trafficability. The moisture content and soil strength data also help in assessing landslide activities.

The electronic block diagram of the STMET depicting all the sensors and their appropriate location in the rod along with the wiring details is shown in Fig 6.

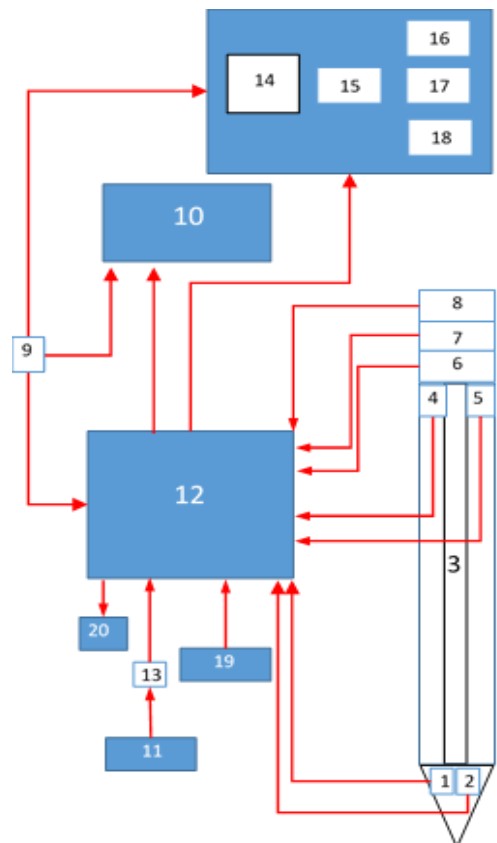

**Figure 6.** Electronics Block Diagram of STMET. Labels are 1. Temperature Sensor 2. Moisture Sensor 3. 6-Pin Transmission Channel 4. Ambient Temperature & Humidity Sensor 5. Depth Sensor 6. Load Cell 7. GPSS+IMU 8. Range Finder 9. Panic Button 10. RF/ Bluetooth Transmitter 11. Power Source 12. Micro-controller 13. Power On/Off 14. Display Unit 15. Hardness Profile 16. Temperature Profile 17. Weak Layer Detection 18. Data Storage 19. USB Port 20. Micro SD Card

Geo-informatics data acquired with the sensors are processed and displayed on the screen. The methodology used for data acquisition is based on CRISP-DM (Cross-Industry standard process for data mining) (Chapman et al., 2000). The data collected through various sensors are processed and stored as shown in the flow diagram (Fig. 7). The onboard 32-bit ARM microcontroller collects data from all sensors and prepares the data packet for onward transmission via a low-power consuming Bluetooth module to ensure seamless data transfer. The data can also be transferred to a PC via a USB port for further analysis.



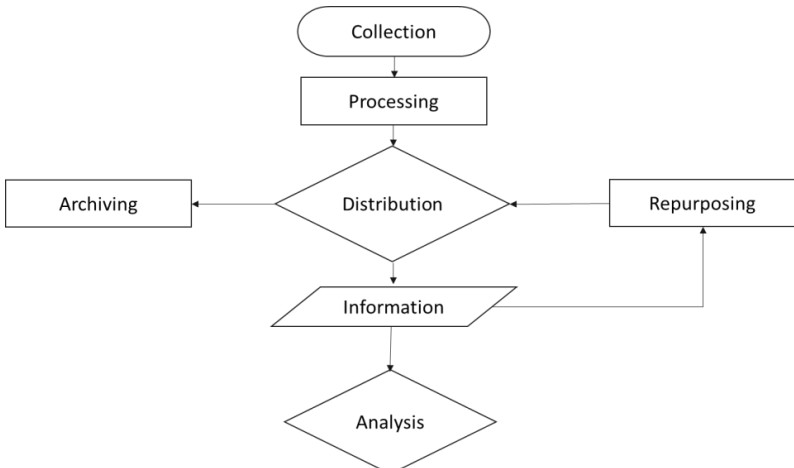

**Figure 7.** Data Acquisition model deployed in STMET

Area Computing Algorithm (ACA) has been implemented in the instrument using Distance range finder and IMU sensors. ACA is based on Gauss's Area formula that relies on the response from laser-based distance measurement and RPY (Roll-pitch-yaw) data of the Inertial measurement unit. The algorithm uses the sensors to locate all the points in the spherical coordinate system and then it transforms them into a Cartesian coordinate system. Then the required area is calculated by Eq. (2):

$$Area = \frac{1}{2}\left\{\begin{vmatrix} x_1 & x_2 \\ y_1 & y_2 \end{vmatrix} + \begin{vmatrix} x_2 & x_3 \\ y_2 & y_3 \end{vmatrix} + \begin{vmatrix} x_3 & x_4 \\ y_3 & y_4 \end{vmatrix} \dots + \begin{vmatrix} x_n & x_1 \\ y_n & y_1 \end{vmatrix}\right\} , \tag{2}$$

Where $(x_1, y_1), (x_2, y_2) \dots (x_n, y_n)$ are the Cartesian coordinates of points located using the instrument and $\begin{vmatrix} x_i & x_j \\ y_i & y_j \end{vmatrix}$ denotes the determinant of the matrix.

The data acquired with the instrument are prompt, maintainable, and evaluable for the contemplation of geohazard. The rescue operation of burial in an avalanche is accelerated using the last known GPS location, and it is further assisted by the Avalanche Victim Detector provided with the instrument separately.

## 4. Results and Discussion

The instrument has been made lighter in weight by using Aluminium alloy and Nylon. These materials have also added features of durability and portability for easy handling in difficult terrain. It is penetrated in snow and soil manually to acquire their physical properties. The penetration rate is kept at 20 – 30 mm/s in the snow so that structure and properties do not get much disturbed and sensors can acquire the required data (Hagenmuller et al., 2016). Snow properties are different in different layers of the snowpack due to metamorphism in the snow, Snowpack structures are changed with diurnal temperature fluctuations and several heterogeneous features like melt-freeze crusts, ice layers, ice columns and snow crystals are evolved within snow. A layer of faceted crystals adjacent to a crust forms a weak layer within the snowpack that affects snowpack stability. The snow strength profile generated with the instrument helps to investigate the weak layer present in the snowpack similar to the manual snow stratigraphy.



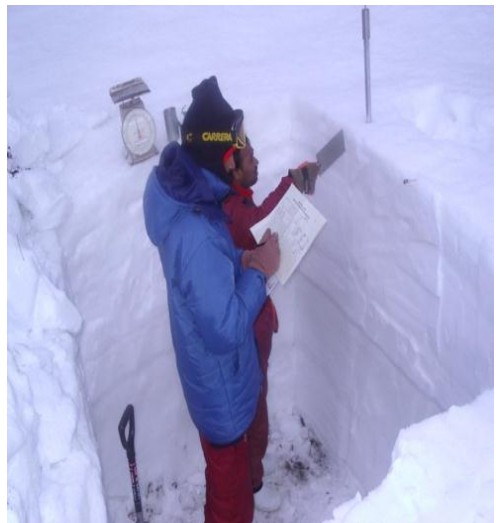

**Figure 8**. Snow pit for Snow Stratigraphy

Figure 8 shows the stratigraphy of the snowpack carried out at an experimental site in the Himalayan region. The manual stratigraphy process is time-consuming (more than 2 hours) having cumbersome processes. The stratigraphy helps researchers to find the instability condition for avalanche forecasting (Reuter et al., 2015). Contrary to traditional stratigraphy, STMET provides strength, moisture and temperature profiles which are the class-2 information for avalanche forecasters (McClung and Peter, 1999) and these are easily collected with STMET. Figure 9(i) shows the comparative hardness profile collected with Ramsonde rod and STMET. It was observed that the hardness data of stratigraphy is almost matching with the data collected with STMET. Similarly, temperature

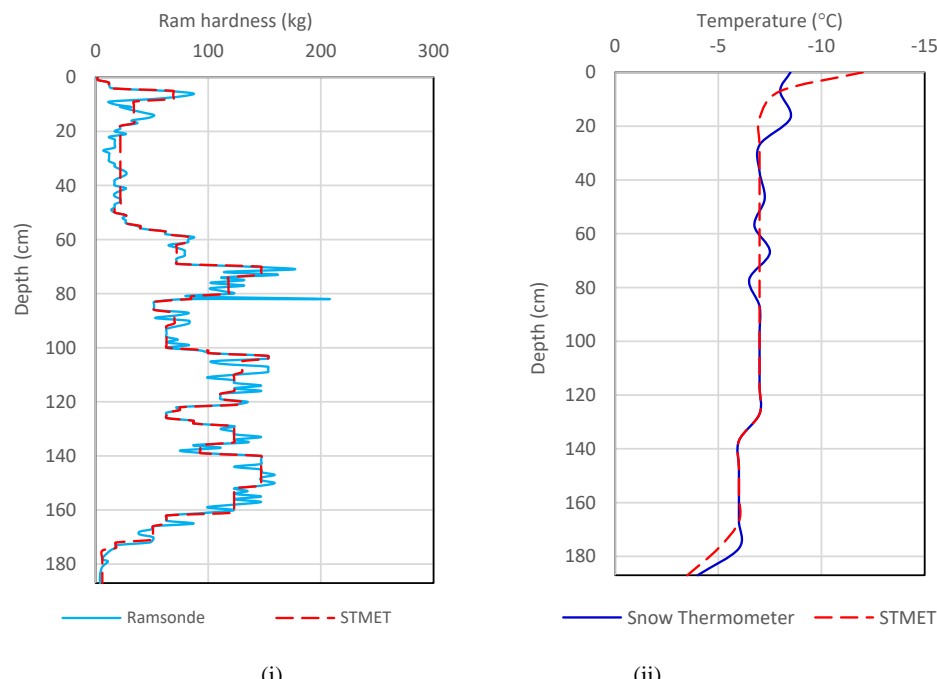

(i)                              (ii)

**Figure 9**. Snow Stratigraphy Data. (i) Snow Strength profile, (ii) Temperature profile



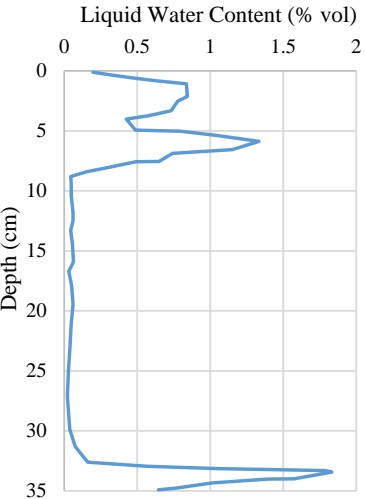

**Figure 10.** Water content in snowpack using STMET

profiles collected with the instrument agree with that of the snow thermometer (Fig.9(ii)). Figure 10 depicts the water content in a snowpack and similar water content data of soil helps in agriculture and assessment of landslides. Acquiring the data with STMET

consumes less time and a visual graph is generated simultaneously to assist decision making in situ.

Similarly, the resistance force of the soil is recorded in terms of the strength of the soil with the instrument. Moisture content and temperature profile of the soil are also obtained simultaneously with STMET. Figure 11 is the graph of the cone index of the soil which represents the penetration resistance of soil. The cone index (CI) parameter is compared with the vehicle cone index (VCI) for the assessment of trafficability. The average values of CI from 0 mm to 150 mm depth are used for comparison with $VCI_1$ for a

single pass of the vehicle and the CI value at 150 mm depth is used for comparison with $VCI_{50}$ (VCI for 50 passes). If the instrument cannot penetrate up to 150mm, extrapolated values of CI up to 150 mm of depth can be used in place of expected values of CI and used for comparison with VCI for trafficability assessment.

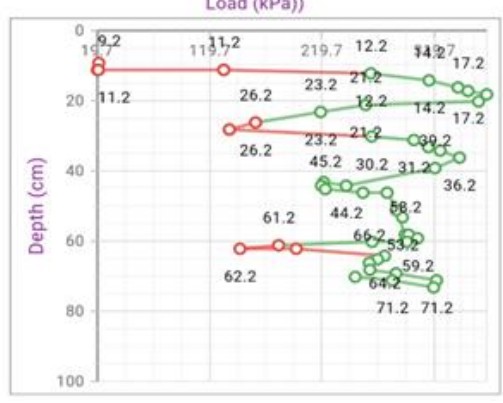

**Figure 11.** Soil Strength using STMET (Screenshot of displayed graph)





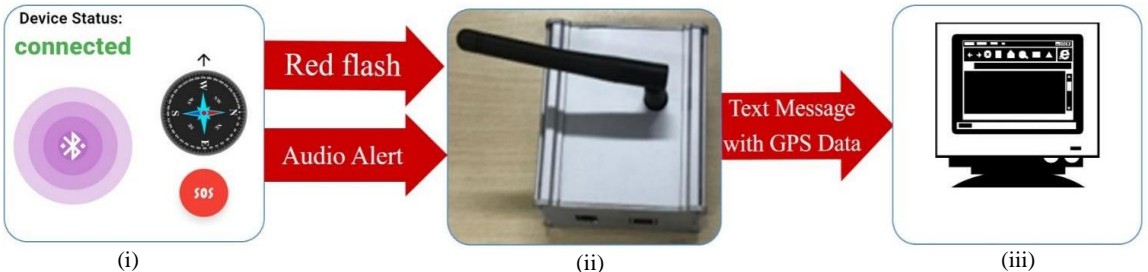

**Figure 12.** SOS/Emergency Response model deployed in STMET. (i) Display with SOS (ii) Receiver module (iii) PC monitor

The transmission module of the STMET provides safety features with a panic button built into it. When the panic button on the electronic box or the display screen is pressed, a signal is transmitted to a base station placed at a distance. The signal alarms the receivers in three modes in terms of sound beeps, flash LED and text message with GPS location which helps to relocate the instrument. The process of transmission of the alert/panic message of the instrument is shown in Fig. 12. The Signal is received by the receiver module which has also an LED bulb to alert the observer. The signal is further processed on the PC and the screen gets

a prompt having GPS location with an alert message. The alert module of the instrument helps in rescue operations to detect the victim who has used the panic button of STMET and also has AVD with transmission mode. Thus, it reduces the exposure time to hazards and the risk involved in difficult terrain.

The Range Finder sensor of the instrument provides distances for planning the movement in difficult areas. An algorithm built into it is used to find the projected area of the formation zone or release area of an avalanche to concisely evaluate the risk. The elevation

angle of the points is determined with the IMU sensor of the instrument for the area measurement as mentioned in section 3.

### 5. Conclusion

Investigation of terrain properties foretells off-road vehicle performance, trafficability, compaction and rutting resulting from vehicle passage in terms of material and property. A similar theory is useful for the assessment of the movement of manpower and baggage in heavy snowfall regions. The cone penetrometer is used for assessment of trafficability and Snow MicroPen has recently

gained prominence for snowpack assessment. Despite all these, there seemed to be an obligation for scientists to have a single instrument that can be resourceful to work in snowy as well as in terrestrial environments and STMET has filled this gap.
The design of the instrument, its material selection and sensors of small form factors have made it portable and easy to operate manually in difficult terrains. The data obtained with the instrument agreed with the stratigraphy data taken by Ramsonde and SMP in the snow field. The moisture parameter of soil is obtained with the instrument along with the strength of the soil. These

parameters are used in the assessment of soil conditions for agriculture and trafficability. The data obtained with the instrument help decision-making and assessment of the stability of snowpacks in avalanche-prone areas and trafficability for vehicle movements respectively. It enhances the safety and rescue of the victim with its alarm system and transmission module. The instrument is designed to enhance the capability of the field operator simultaneously reducing the exposure to the risk during snowpack stratigraphy. The instrument is packed with high-resolution sensors and accurate telemetry so that it can assist the

operator with prompt decision-making. The multi-functional features of the STMET will certainly purge the hauling of separate



equipment during the movement in the difficult snow-bound terrain helping in easy mobility and collecting snow parameters for avalanche defence.

## 6. Author contributions

GK conceptualized the system, designed the mechanical components, and developed the 3D CAD model for fabrication, based on his field experience. ZS identified suitable materials, supervised the fabrication process, and assisted with sensor development. He also contributed to the development of the software and algorithms. Both authors collaborated on planning the experiments, deriving models, and analyzing the data. All authors contributed to the interpretation of results and the creation of figures. GK led the manuscript writing and incorporated feedback from colleagues and reviewers.

## 7. Competing interests

The contact author has declared that none of the authors has any competing interests.

## 8. Acknowledgement

We express our thanks and appreciation to M. K. Kalra for his constant encouragement and valuable comments during the development of the Snow and Terrain Mobility Evaluation Tool (STMET). The authors also extend their gratitude to Upika Mittal for her contribution to the development of electronic sensors and the drafting of the paper.  The instrument has been developed
under a project funded by DRDO (DGRE), India. Hence, the Director, DGRE and Project Director (M. K. Kalra) are acknowledged for their support. Scientists and technical staff of DGRE have also contributed to the testing and trial of the instrument in the field and laboratory. We are also grateful to M/s New Age Instruments and Material for technical support of its fabrication.

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
