# Peer review of "Geo-Informatics Data Acquisition Instrument for Assessment of Avalanche Risk and Trafficability: STMET"

_EGUsphere, 2024_

## Author Comment (AC3)

**RC2:**'Comment on egusphere-2024-3154', *Anonymous Referee #2*, 06 May 2025

The overall organization of the article is clear and meets the requirements for writing scientific papers. Off-road conditions and snowpack strata data are traditionally acquired with Cone penetrometer and Ramsonde Rod to assess traditionally acquired with Cone penetrometer and Ramsonde Rod to assess trafficability and snow stability. On this basis, the author designed a new Snow and Terrain Mobility Evaluation Tool (STMET) and applied it. I believe that this article can be accepted after providing additional technical details and revisions.

**Response to Referee 2:**

The authors sincerely thank the referee for the positive and encouraging comments. The newly developed STMET instrument offers several advantages over existing tools, such as the cone penetrometer (used in soil analysis) and the Ramsonde rod (used in snowpack evaluation). In addition to the functionalities of these instruments, STMET is equipped with additional features, including the capability to measure temperature profiles and moisture content-features absent in both traditional tools.

They have carefully reviewed and revised the manuscript by incorporating all feedback and suggestions provided by the esteemed referees. The updated manuscript is submitted with the hope that it meets the standards for publication.

Point-wise responses to the referee's comments are as follows:

1.What is the condition of the new Snow and Terrain Mobility Evaluation Tool (STMET) battery and how long can it be used in the field?

**Response 1:** The STMET is powered by a 6000 mAh lithium-ion cylindrical battery, which is housed within the handle of the device. Under standard operating conditions, the battery provides a backup of approximately 4 hours.

2. The author mentioned the Inertial Measurement Unit (IMU), how do they work, and what contribution do they make to the final measurement results?

**Response 2:** The IMU comprises an accelerometer, gyroscope, and magnetometer. It measures the acceleration, velocity, and orientation (pitch, roll, and yaw) of the device. This information is used to determine the penetration rate of STMET for optimal operation. Additionally, the orientation data is integrated into an algorithm that computes terrain area and slope, contributing to a better understanding of terrain features.

3. The article discusses the measurement results of temperature and humidity, but lacks modeling based on these parameters and terrain passability. How will these measurement results be applied?

**Response 3:** The authors thank the referee for this valuable observation. The STMET includes temperature and humidity sensors mounted on the electronic enclosure to monitor ambient environmental conditions. In emergencies, this data can be transmitted to a base station upon activation of a panic button.

To evaluate terrain trafficability (passability), additional temperature and moisture sensors are embedded at the rod's tip to measure subsurface conditions. Soil strength is influenced by its

moisture content and is quantified using the cone index. This is compared with the Vehicle Cone Index (VCI) of a specific vehicle to determine the feasibility of terrain traversal. The methodology has been elaborated in the revised manuscript.

4. The article mentions The Decision Support System (DSS), which decisions can ultimately be made by the system, and the decision-making process should be supplemented and explained.

**Response 4:** Authors appreciate the referee's insightful suggestion. The DSS integrated with STMET aids users in decision-making across the following applications:

1. **Weak Layer Identification:** Based on the measured strength profile, users can detect weak layers within the snowpack, which are potential triggers for snow avalanches.
2. **Terrain Slope Assessment:** The instrument measures terrain slope, which is essential for classifying avalanche-prone zones—formation, middle, and runout.
3. **Trafficability Assessment:** An algorithm uses measured soil strength and cone index data to determine whether a specific vehicle can traverse a given terrain and estimate the number of feasible passes.
4. **Victim Detection and Rescue Support:** In the event of an accident, STMET can send an alert signal containing GPS coordinates and ambient temperature to facilitate timely rescue operations.

These details have been included in the revised manuscript.

5. The references should be further supplemented.

**Response 5:** The reference list has been reviewed and updated. Duplicate entries have been removed, and additional relevant references have been included to support the discussion.

---

## Author Response (AR2)

**RC1**: 'Comment on egusphere-2024-3154', Anonymous Referee #1, 22 Apr 2025 reply

The manuscript "Geo-informatics data acquisition instrument for assessment of avalanche risk and trafficability: STMET" is an interesting read. The work presented is quite far from my own area of research, so I cannot adequately comment on the intended application or the novelty. My main issue with the manuscript is a confusing structure where background information, technical design choices, results and discussions are mixed up. The manuscript would greatly benefit from a major revision focusing on structure. One example is the start of the results section which opens with the choice of materials for the instrument and then proceeds to introduce snow properties. These bits of information should be moved to appropriate sections. Another example is the dual use of the instrument in snow and soil. A clearer distinction between the different use cases would be appreciated.

**Authors' Response:**

The authors are sincerely thankful to the referee for the thoughtful and constructive comments on our manuscript titled *"Geo-informatics data acquisition instrument for assessment of avalanche risk and trafficability: STMET"*. We appreciate the time and effort invested in reviewing our work. We have carefully considered each comment and revised the manuscript accordingly to improve its technical clarity, organization, and overall presentation. We have incorporated all suggestions in the modified manuscript to meet the standard of the journal publication.

Authors' responses are given below point-by-point to the referee's specific comments.

**Comment from Reviewer:**

Line 95: It is not clear to me how the laser range finder works. According to the text it is placed inside the box. Please clarify and potentially revise Figure 1 to make this clear.

**Authors' response:** The authors agree with the referee that this section required greater clarity. The laser range finder and depth measuring sensors operate on the principle of Time of Flight (ToF). On line 95, the depth measuring system has been explained and it is mounted beneath the electronics box, pointing downward through an aperture in the base of the box. All sensors could not be shown in Figure 1 due to visual complexity and so they have illustrated the location and orientation in the figure. The depth measuring sensor is separately shown in Figure 3 for more clarity. The corresponding text has also been revised for clarity.

**Comment from Reviewer:**

Line 113: It is claimed that the time response of the temperature sensor is fast. Unfortunately, there is no definition of "fast", nor is any data presented to warrant this claim.

**Authors' response:** The authors appreciate the referee's point. The manuscript has been revised to include the quantitative response time of the temperature sensor. The response time is 0.25 s which meets the requirements for rapid surface and subsurface temperature acquisition in dynamic environments.

**Comment from Reviewer:**

Line 128: A calibration is mentioned, but there is no discussion of the precision and reproducibility of this sensor.

**Authors' response:** The authors are thankful to the referee for this important remark. We have described the procedure and precision in the section of the modified manuscript. The moisture sensor was calibrated using distilled water (100% moisture reference) and ambient air (0% reference). Subsequent tests involved adding measured volumes of water to known soil samples. Each calibration scenario was repeated ten times by different operators to assess reproducibility. The maximum deviation in repeated measurements was within ±2%. These details have been included in the revised manuscript.

**Comment from Reviewer:**

Line 138 (and comment on line 160 below): I suggest adding a few details. Are data manually recorded?

**Authors' response:** We agree that this needed clarification. We have now added a detailed explanation of the data acquisition process. All sensor data are recorded manually.

**Comment from Reviewer:**

Figure 6: This could be made much easier to follow if the names of parts appeared in the figure instead of being moved to the caption.

**Authors' response:** We appreciate the suggestion for the naming of parts of the instruments. While we initially attempted to label parts within the figure, the density and overlap of components made the figure visually cluttered and difficult to interpret. Therefore, we opted to retain the original caption-based naming system, which maintains figure clarity. This format was also acceptable to other referees.

**Comment from Reviewer:**

Figure 7: This figure does not contribute much to the paper and can be removed.

**Authors' response:** We respectfully differ on this point. Figure 7 illustrates the overall methodology and data acquisition process, which complements the textual explanation and enhances understanding of the data workflow. We believe this figure adds value for readers and have therefore retained it in the revised manuscript.

**Comment from Reviewer:**

Line 160: The area computing algorithm is rather abruptly placed in the manuscript. I suggest adding a few sentences on the background of it. Further, the distance range finder is poorly described. Is it just a separate instrument or is it part of the new design?

**Authors' response:** We appreciate this observation. The area computing algorithm has been mentioned in the distance measuring section 2.2.6, now to implement the feedback. Additional context has been provided to explain the area computing algorithm, which was developed to measure terrain slope area using the laser range finder and the integrated IMU. The laser range finder is a component of the STMET instrument, not a separate instrument. These details have been clarified and the related section has been revised accordingly.

**Comment from Reviewer:**

Line 192: An example of a profile of moisture and temperature in soil, like figures 9 and 10 would be appropriate. Where is the comparison between CI and VCI? What does the colors in Figure 11 indicate?

**Authors' response:** We are thankful to the referee for these valuable suggestions. We have added a soil profile showing moisture and temperature to the manuscript. Additionally, we have clarified that the Cone Index (CI) measured by the instrument is compared with the Vehicle Cone Index (VCI) to assess terrain trafficability. The color scheme used in Figure 11 was solely for visual differentiation and does not carry any specific meaning.

**Comment from Reviewer:**

Technical corrections:

Line 261: The McClung and Schaerer reference appears twice. Please remove one.

**Authors' response:** We appreciate the referee's observation over references. The duplicate reference has been removed.

**RC2:** 'Comment on egusphere-2024-3154', *Anonymous Referee #2*, 06 May 2025

The overall organization of the article is clear and meets the requirements for writing scientific papers. Off-road conditions and snowpack strata data are traditionally acquired with Cone penetrometers and Ramsonde Rod to assess traditionally acquired with Cone penetrometers and Ramsonde Rod to assess trafficability and snow stability. On this basis, the author designed a new Snow and Terrain Mobility Evaluation Tool (STMET) and applied it. I believe that this article can be accepted after providing additional technical details and revisions.

**Authors' response to Referee 2:**

We sincerely thank the referee for the positive and encouraging comments. The newly developed STMET instrument offers several advantages over existing tools, such as the cone penetrometer (used in soil analysis) and the Ramsonde rod (used in snowpack evaluation). In addition to the functionalities of these instruments, STMET is equipped with additional features, including the capability to measure temperature profiles and moisture content-features absent in both traditional tools.

We have carefully reviewed and revised the manuscript by incorporating all feedback and suggestions provided by the esteemed referees. The updated manuscript is submitted with the hope that it meets the standards for publication.

Point-wise responses to the referee's comments are as follows:

**Comment from Reviewer:**

1. What is the condition of the new Snow and Terrain Mobility Evaluation Tool (STMET) battery and how long can it be used in the field?

**Authors' response 1:** The STMET is powered by a 6000 mAh lithium-ion cylindrical battery, which is housed within the handle of the device. Under standard operating conditions, the battery provides a backup of approximately 4 hours.

**Comment from Reviewer:**

2. The author mentioned the Inertial Measurement Unit (IMU), how do they work, and what contribution do they make to the final measurement results?

**Authors' response 2:** The IMU comprises an accelerometer, gyroscope, and magnetometer. It measures the acceleration, velocity, and orientation (pitch, roll, and yaw) of the device. This information is used to determine the penetration rate of STMET for optimal operation. Additionally, the orientation data is integrated into an algorithm that computes terrain area and slope, contributing to a better understanding of terrain features.

**Comment from Reviewer:**

3. The article discusses the measurement results of temperature and humidity but lacks modeling based on these parameters and terrain passability. How will these measurement results be applied?

**Authors' response 3:** The authors thank the referee for this valuable observation. The STMET includes temperature and humidity sensors mounted on the electronic enclosure to monitor ambient environmental conditions. In emergencies, this data can be transmitted to a base station upon activation of a panic button.

To evaluate terrain trafficability (passability), additional temperature and moisture sensors are embedded at the rod's tip to measure subsurface conditions. Soil strength is influenced by its moisture content and is quantified using the cone index. This is compared with the Vehicle Cone Index (VCI) of a specific vehicle to determine the feasibility of terrain traversal. The methodology has been elaborated in the revised manuscript.

**Comment from Reviewer:**

4. The article mentions The Decision Support System (DSS), which decisions can ultimately be made by the system, and the decision-making process should be supplemented and explained.

**Authors' response 4:** Authors appreciate the referee's insightful suggestion. The DSS integrated with STMET aids users in decision-making across the following applications:

1. **Weak Layer Identification:** Based on the measured strength profile, users can detect and locate weak layers within the snowpack, which are potential triggers for snow avalanches.
2. **Terrain Slope Assessment:** The instrument measures the slope of the terrain, which is essential for classifying avalanche-prone zones—formation, middle, and runout.
3. **Trafficability Assessment:** An algorithm uses measured soil strength and cone index data to determine whether a specific vehicle can traverse a given terrain and estimate the number of feasible passes.
4. **Victim Detection and Rescue Support:** In the event of an accident, STMET can send an alert signal containing GPS coordinates and ambient temperature to facilitate timely rescue operations.

These details have been included in the revised manuscript.

**Comment from Reviewer:**

5. The references should be further supplemented.

**Authors' response 5:** The reference list has been reviewed and updated. Duplicate entries have been removed, and additional relevant references have been included to support the discussion.